# Gut–Kidney Axis Investigations in Animal Models of Chronic Kidney Disease

**DOI:** 10.3390/toxins14090626

**Published:** 2022-09-07

**Authors:** Piotr Bartochowski, Nathalie Gayrard, Stéphanie Bornes, Céline Druart, Angel Argilés, Magali Cordaillat-Simmons, Flore Duranton

**Affiliations:** 1RD Néphrologie SAS, 34090 Montpellier, France; 2BC2M, Faculty of Pharmacy, University of Montpellier, 34090 Montpellier, France; 3Université Clermont Auvergne, Inrae, Vetagro Sup, UMRF0545, 15000 Aurillac, France; 4Pharmabiotic Research Institute (PRI), 11100 Narbonne, France

**Keywords:** microbiota, CKD, animal models

## Abstract

Chronic kidney disease (CKD) is an incurable disease in which renal function gradually declines, resulting in no noticeable symptoms during the early stages and a life-threatening disorder in the latest stage. The changes that accompany renal failure are likely to influence the gut microbiota, or the ecosystem of micro-organisms resident in the intestine. Altered gut microbiota can display metabolic changes and become harmful to the host. To study the gut–kidney axis in vivo, animal models should ideally reproduce the disorders affecting both the host and the gut microbiota. Murine models of CKD, but not dog, manifest slowed gut transit, similarly to patient. Animal models of CKD also reproduce altered intestinal barrier function, as well as the resulting leaky gut syndrome and bacterial translocation. CKD animal models replicate metabolic but not compositional changes in the gut microbiota. Researchers investigating the gut–kidney axis should pay attention to the selection of the animal model (disease induction method, species) and the setting of the experimental design (control group, sterilization method, individually ventilated cages) that have been shown to influence gut microbiota.

## 1. Introduction

Chronic kidney disease (CKD) is defined as abnormalities in kidney structure or function that last over 3 months and have health implications [1]. In CKD, the decrease in renal function leads to the accumulation of various compounds in the blood, known as uremic solutes. Increased blood concentrations of uremic solutes promote uremic syndrome, which comprises many organ disorders and cellular dysfunctions [2,3]. CKD patients have been reported to display reduced gut motility, increased intestinal permeability, bacterial overgrowth, bacterial translocation, and inflammation of intestinal origin [4,5,6], which are consistent with dysbiosis. Although the main cause of intestinal dysbiosis in CKD patients remains unknown, there is evidence that a low-fiber [7], low-protein diet [8], medications and treatments [9,10,11], and reduced renal function alter the biochemical environment of the gut and modify the gut microbiota (GM), or the ecosystem of micro-organisms inhabiting the intestine. Uremic syndrome affects the gastrointestinal tract with manifestations including anorexia, nausea, ulcers, malnutrition, and protein-energy wasting [12]. CKD may favor the growth of uremic toxin-producing bacteria. In parallel, the reduced abundance of carbohydrate-fermenting bacteria leads to the decreased production of beneficial short-chain fatty acids (SCFAs) [13], which are involved in the regulation of glycemic and lipid metabolism, immune system response, blood pressure, and intestinal barrier functions [14]. Alterations in the gut–kidney axis (Figure 1) may adversely affect the development of CKD and increase the risk of comorbidities, such as cardiovascular disease [15], which is the leading cause of mortality in patients with CKD.

With the ultimate goals of prevention and treatment in mind, changes in the gut and GM deserve further characterization, for which in vivo studies are needed. To adequately study the gut–kidney axis, animal models of CKD and experimental setups should meet the requirements specific to GM research and adequately reproduce the disorders affecting the host and GM. In the present review, we will evaluate how animal models of CKD perform with regard to changes in the gut and GM and discuss the validity of the models.

## 2. Materials and Methods

In order to test the validity of using animal models of CKD to study alterations to the gut–kidney axis, an electronic search was performed on material published from 1 January 1990 to 1 April 2022, in PubMed. We searched combinations of the terms: “chronic kidney disease”, “chronic renal disease”, “chronic renal failure”, “gut microbiota”, “gut microbiome”, “intestinal microbiota”, “tight junction”, “gut barrier”, “gut barrier transport”, “gut motility,” “intestinal transit,” “leaky gut”, “gut hyperpermeability”, “bacterial translocation”, and “endotoxemia”. Search results were limited to English-language publications. In addition, references of selected studies were screened to identify eligible trials that were not retrieved by database searches. We included studies on animal models of CKD only. Each search result was independently reviewed for eligibility by the author (P.B.). We excluded trials that reported: (1) no details on CKD induction in animal models; (2) no results on intestinal disturbances, and (3), for articles showing gut microbiota alterations, no indication of the effect of compositional changes. We retrieved 23 studies (Table 1).

## 3. CKD Models Used to Assess the Gut–Kidney Axis

A variety of animal models were used to explore the pathophysiology of CKD and gut-specific alterations (Table 1). Although rodent models of CKD are predominant, other species are used, including dogs [16]. Furthermore, a range of CKD induction methods is available to mimic desirable aspects of the disease, such as a CKD model induced by an adenine-enriched diet. Adenine is metabolized to uric acid [39] and 2,8-dihydroxyadenine [40], which accumulate as tubular-shaped crystals in the kidney, causing damage to its parenchyma [40], leading to fibrosis and, ultimately, uremia [41]. Surgical models have been developed as an alternative to diet-based models. A subtotal five-sixths nephrectomy (SNx) generally consists of the removal of one kidney and two-thirds of the contralateral. Renal mass reduction results in increasing the single-nephron glomerular filtration rate in the remaining nephrons, which subsequently leads to their injury and a gradual decline in renal function [42]. Unilateral ureteral obstruction (UUO) is another surgical model in which the ureter is durably or transiently ligated. Obstructed urinary flow results in upstream changes, mechanical stretching, and cell apoptosis [43]. These changes are related to tubular cell damage, kidney inflammation, and fibrosis [44]. UUO may model the transition from acute renal failure to CKD.

## 4. Motility Dysfunction and Gut Barrier

The intestinal barrier plays an important role in the host–GM relationship, as it separates the gut lumen environment from the host organism. The intestinal barrier passively and actively regulates nutrients’ absorption and prevents the transfer of harmful substances into the bloodstream [45]. Gut motility is essential for the progression of food down the gastrointestinal tract during digestion, absorption of nutrients, regulation of bacterial growth, and removal of undigested and harmful compounds [5].

The manometric examination of the bowel in patients with CKD revealed abnormal motility patterns in the small intestine, which correlated with a higher risk of bacterial overgrowth [5]. Colon motility, aside from small intestine motility, might be impaired in CKD. The colonic transit time increased from 24 ± 12 h in healthy subjects to 33 ± 14 h in continuous ambulatory peritoneal dialysis patients and 43 ± 22 h in hemodialysis patients [46]. Elongated intestinal transit was observed in the SNx rat model [17,18,19] and the mouse model induced by an adenine-enriched diet [20]. In contrast, oro-anal transit and colonic transit times were decreased in SNx dogs [16]. The effect of the dialysis modality on gut motility in animal models was not assessed.

Leaky gut, or intestinal hyperpermeability, is a phenomenon that occurs when the structure of tight junctions and/or transcellular transport is impaired, enabling harmful substances to enter the bloodstream [47]. It can be detected by the presence in the blood of significant amounts of molecules normally present in the gut lumen but absent in blood (unable to cross the gut barrier), such as diamine-oxidase. Diamine-oxidase has been found to be present in the blood of both CKD and end-stage renal disease (ESRD) patients, whilst it is not observed in healthy subjects [48]. Using orally administered markers (sucrose, sucralose, or lactulose/mannitol ratio) and measuring urinary recovery, a higher colon permeability has been shown in CKD patients compared to healthy individuals [4]. The administration of a symbiotic formula containing prebiotics, probiotics, and antioxidants reduced small intestine permeability in CKD patients [4]. In animal studies, gut permeability is usually determined by the level of circulating endotoxins, mainly lipopolysaccharides (LPS). In uremic rats, serum LPS levels were greatly increased compared to controls [21]. However, LPS is not an ideal marker of leaky gut, as it can be produced by blood bacteria in the case of infection [49]. Using orally administered exogenous markers (e.g., FITC-dextran) is a more precise alternative that has confirmed the higher gut leakage in SNx mice compared to sham [22,23].

One of the consequences of a leaky gut is greater bacterial translocation, which is defined as the passage of viable bacteria, bacterial cell elements, and bacterial products through the gut barrier to normally sterile tissues, such as blood, mesenteric lymph nodes, and internal organs [50]. In two studies, about 20% of ESRD patients were found to have bacteria-derived DNA fragments in their blood, which is a sign of bacterial translocation [6,51]. The bacterial DNA found in the blood of ESRD patients belonged to the taxa *Escherichia coli*, *Staphylococcus aureus*, *Pseudomonas aeruginosa*, *Staphylococcus epidermis*, *Enterococcus faecalis*, *Proteus mirabilis*, *Staphylococcus haemolyticus* [51], and *Klebsiella* [6]. Translocation of bacterial DNA was correlated with inflammatory markers, such as serum IL-6 and C-reactive protein [6,51]. In animal models, bacterial translocation can be seen in rats with CKD induced by an adenine-enriched diet [24] and by SNx [25,26]. Interestingly, in SNx rats, translocated bacterial DNA showed similarity in taxa to those found in humans. The frequency of translocations was much higher in animal models than in humans [25]. The passage of viable bacteria from the gastrointestinal tract to extra-intestinal sites, including blood, was reported in SNx rats [26]; however, this was not confirmed in another study [25], and no viable bacteria were found in hemocultures of CKD patients [51].

The intestinal barrier consists of a monolayer of tightly sealed epithelial cells covered at their apical poles by a mucus layer produced by goblet cells [45]. Tight junctions strengthen the cell-to-cell contact, limiting the paracellular route for penetration of molecules into the organism [45]. In a small cohort of CKD patients, the histological examination of tight junction proteins in the colon showed a moderate reduction in occludin, while the zo-1, claudin-1, and claudin-4 proteins were unchanged [52]. In hemodialyzed patients, a decrease in claudin-1 and zo-1 proteins was observed in the histological analysis, and the reduction in occludin was greater compared to that in CKD patients [52]. Changes in the abundance of tight junction proteins in the colon and other parts of the intestine [27] were observed in CKD rats [21,24,27,28,29,30,31,32] and mice [22,23,33], regardless of the method of induction [29]. The abundance of tight junction proteins was reduced [21,23,24,27,28,29,30,31,32,33] and structural alterations of tight junctions, such as reduced density and widened intercellular space, were observed [22]. Interestingly, mRNA expressions of tight junction components were decreased in the colon of SNx mice [22] and increased in SNx rats despite lower protein levels [29]. Although it seems that protein level and gene expression should be positively associated, they are actually hardly comparable due to the post-transcriptional mechanisms involved in the conversion of mRNA to protein, the different half-lives of mRNAs and proteins in vivo, and, finally, the different technical properties of detection methods of protein and mRNA make them difficult to compare [53]. Reduced concentrations of SCFAs (products of commensal bacteria) in serum and feces have been reported in CKD patients compared to controls [54]. In animal models of CKD, SCFAs levels were also reduced [34,35]. In SNx rats, supplementation with the SCFA butyrate restored the number of tight junction proteins [21]. Similar effects were observed after supplementation with non-digestible carbohydrates [33,36] that are fermented by the GM, resulting in SCFAs production. The effects of SCFAs or indigestible carbohydrates’ supplementation on tight junction protein abundance have not been examined in CKD patients.

Interestingly, the mucus layer has been reported to be altered in CKD animals. A reduced expression of mucin-2 (a main component of mucus) and lower mucus production were found [21]. Mucus plays the dual role of reducing contact between the GM and gut epithelial cells and providing a source of nutrients for bacteria [55]. To our knowledge, potential changes in mucus secretion and composition in CKD patients have not been reported.

Considerably less is known about changes in the transcellular pathway in CKD. The transcellular route depends on a complex network of channels and transporters regulated by sensing and signaling machinery [56]. Chronic changes in the biochemical milieu associated with renal disease may result in dysfunction of the transcellular route [56]. ABCG2, a urate transporter present in both kidneys and intestines, appears to play an important role in CKD. In ESRD patients, a group with dysfunctional variants of the ABCG2 gene had reduced ABCG2-mediated intestinal urate excretion resulting in increased hyperuricemia [57]. Increased expression of the ABCG2 transporter in the ileum was observed in SNx rats [36], suggesting that intestinal urate excretion may increase with renal insufficiency.

Overall, CKD appears to be associated with an impairment of paracellular and transcellular transport, which contributes to the phenomenon of leaky gut, bacterial translocation, and associated inflammation of gut origin.

## 5. Gut Microbiota Alterations in CKD

CKD may also modify the composition of the GM. In ESRD patients, the expansion of proteolytic bacteria involved in uremic toxins’ metabolism was observed [13]. Among 19 overgrowing bacterial families, 12 possessed the urease gene (*Alteromonadaceae, Cellulomonadaceae, Clostridiaceae, Dermabacteraceae, Enterobacteriaceae, Halomonadaceae, Methylococcaceae, Micrococcaceae, Moraxellaceae, Polyangiaceae, Pseudomonadaceae*, and *Xanthomonadaceae*), and 5 harbored the uricase gene (*Cellulomonadaceae, Dermabacteraceaea, Micrococcaceae, Polyangiaceae*, and *Xanthomonadaceae*) producing ammonia that is harmful to the intestinal epithelium. In addition, families expressing the tryptophanase gene (*Clostridiaceae, Enterobacteriaceae*, and *Verrucomicrobiaceaea*), necessary for the conversion of tryptophan into indole, which is then metabolized by the liver into uremic toxins, were also overrepresented. *Clostridiaceae* and *Enterobacteriaceae* also possess enzymes that produce p-cresyl from tyrosine. In contrast, the relative abundance of carbohydrate-fermenting bacteria (*Lactobacillacease* and *Prevotellaceae* families) decreased in ESRD patients [13]. In CKD and ESRD patients, the reduced concentrations of SCFAs in blood and stool samples were associated with a lower abundance of *Enterobacter, Enterococcus, Bifidobacterium, Bacteroides, Clostridium, Faecalibacterium*, and *Roseburia,* most of which are SCFAs-producing taxa [54]. Interestingly, the SCFAs’ concentration in the blood was inversely correlated with the degree of renal insufficiency in patients [54].

Due to the different genetic backgrounds of the host and environmental conditions, it might be difficult to reproduce compositional changes in animal models. The compositional diversity of the GM varied according to the method of induction: it increased in the SNx model [37], did not change in the UUO model [34], and decreased in the adenine-enriched diet [35], while it mostly decreased in CKD and ESRD patients [58]. In the UUO rat model, bacteria belonging to the taxa *Acetatifactor, Blautia, Intestinimonas*, and *Oscillibacter* were more abundant, while the taxa *Lactobacillales, Clostridium, Streptococcus*, and *Pseudomonas* were reduced [34]. A decrease in the plasma tryptophan level and an increase in tryptophan metabolites were associated with a lower abundance of *Clostridium, Pseudomonas*, and *Lactobacillales*, which hold the tryptophan synthase gene, and a higher abundance of *Oscilibacter, Blautia*, and *Intestinimonas*, which are able to catabolize tryptophan into indoles [34]. In adenine rats, changes in the GM were associated with an increase in the trimethylamine N-oxide (TMAO) blood level [35]. GM changes included a reduced abundance of the carbohydrate-fermenting bacteria *Bacteroidetes* and an increase in *Lachnospiraceae, Erysipelatoclostridium, Tannerellaceae, Akkermansiaceae, Verrucomicrobiae, Oscillibacter, Flavonifractor*, and *Parabacteroides,* which were positively associated with TMAO levels [35]. In SNx rats, the GM composition was modified with an increased abundance of *Methanosphaera, Akkermansia, Christensenella, Adlercreutzia, Corynebacterium*, and *Coprobacillus*, as well as genera from the class *Mollicutes* and family *Clostridiacea,* which was correlated with amino acids’ metabolism [37].

Interestingly, the animal models display changes in the composition of GM that result in metabolic changes comparable to clinical observations. Altogether, animal models of CKD manifest changes in the GM composition and function that are consistent with the increased production of uremic toxins and decreased production of SCFAs, as in humans.

Overall, the alterations in the gut structure and functions, as well as in the GM, that have been reported in CKD patients appear to be reproduced, at least partly, in animal models designed to reproduce CKD. For observations from animal models that have not been shown in humans, it is highly relevant to evaluate the validity of the model and experiment to assess how representative of the human disease they are, and, therefore, how the results can be extrapolated from preclinical models to humans.

## 6. Adaptation of CKD Animal Models to GM–Kidney Research

The value of an animal experiment is defined by the accuracy of the translation of the results into clinical research [59]. Successful translation is based on proper experiment design and the selection of an adequate animal model [59]. Internal validity addresses the consistency of the experimental design and its reproducibility [60]. To minimize the researcher’s influence on the obtained results, randomization techniques and blind testing may be applied [59]. A meta-analysis of 290 animal studies showed that experiments with randomization and/or blind-testing were more likely to report reproducible results [61]. These methods can be implemented in the research on the gut–kidney axis to avoid biases, although disease induction is often associated with important phenotypic changes, thereby limiting the blinding of model induction.

External validity, or how the results obtained in the preclinical models can be translated into a better understanding and predictability of human conditions, is also important [59]. When designing animal models, the first thing that needs to be taken into account is the animal species. To mimic human CKD, various species can be used that differ in kidney structure, body size, blood volume, and genetics, which impact the choice of CKD induction methods, the similarity of CKD disease progression, and the choice of analytical techniques [62]. The specificity of GM investigation also requires considering the anatomical and physiological characteristics of the animal used for the study. The similarity of the intestinal tract as an environment of GM between animals and humans is a prerequisite. Mammals manifest great functional and anatomical similarities between orders, though species variation needs to be acknowledged. Among them, differences in the length and volume of parts of the gastrointestinal tract, species-specific intestinal wall structures [63], variable pH, and luminal enzyme secretions are observed [64]. As the digestive tract strongly reflects the adaptation of the species to the diet, omnivorous animals, such as rats, mice, pigs, or monkeys, are preferable. Furthermore, the choice of omnivorous animals allows the implementation of a wider spectrum of dietary interventions. Other limitations include the physical capacities of the gastrointestinal tract that exist in humans but not in other species, such as the ability to vomit (absent from rodents) [65]. Another major obstacle is behaviors and eating habits manifested by animals and not by humans, such as coprophagy, which has a strong impact on GM composition but cannot be prevented without serious health implications [66].

Laboratory animal breeders provide specific pathogen-free (SPF) animals that are free of commonly occurring pathogens and parasites and free of the corresponding diseases. Due to their genetic homogeneity and similar living environment, results obtained on colony animals are relatively easy to replicate. However, the low genetic diversity does not reflect the broader and more varied population of CKD patients. For these reasons, the idea of “dirty” animal models may better represent human populations, where genetic differences, contact with pathogens, and living environment shape GM composition at the expense of a lower control [67]. The use of pet animals with greater genetic diversity, different ages, and health conditions would better reflect an unselected population. Still, it remains controversial, as uncontrolled and unknown factors shaping the GM composition may bias the results [67]. The method of induction of CKD also needs to be thought upon. In GM research, methods of CKD induction are based on surgery or diet modifications (Table 2). The changes in GM composition reported by an adenine-enriched diet are variable, ranging from clearly observed [68] to minimal [38] or absent [69]. However, animals fed an adenine-rich diet have significantly lower body masses [38,68,69]. A lower food intake is also observed, probably due to both CKD and diet aversion [70], which result in vitamin deficiency and malnutrition [38] and can directly cause shifts in the GM composition. A major limitation of the adenine-enriched diet-induced model is the impossibility of setting up a control group to distinguish the effect of diet on GM from the CKD–GM interactions. In surgical models, the use of drugs, such as antibiotics [71], anesthetics [72], and analgesics [73], can affect the GM composition up to several weeks after surgery. Moreover, in some studies, it has been observed that the body weight of operated animals is lower compared to control animals [27]. Still, sham surgery can help distinguish the effects of surgery and drugs on GM from those of CKD.

Special attention should also be paid to the animal environment. Diet shapes the GM composition [75], and GM mediates nutrient absorption and host metabolism [76]. To avoid fluctuations in GM composition, commercially available diets with a fixed composition should be considered. The quality of drinking water must also be maintained constant during the experiment. In particular, the presence of residual micro-organisms (usually from *Proteobacteria*, *Bacterioidetes*, or *Actinobacteria*, such as *Acinetobacter*, *Pseudomonas*, and *Mycobacterium*) [77], the pH of the water [78], and the concentration and ion ratio [78,79] may also influence the GM composition. For rodents, the effect of bedding and supplementation with fiber-rich wooden bricks or nests should also be evaluated as their consumption leads to reduced nutrient absorption [80], growth of fiber-fermenting bacteria, and increased production of metabolites, such as SCFAs [81]. Bedding consumption can be reduced by *ad libitum* feeding [81]. Diet, water, and bedding can be sterilized to decrease the inoculation of animals with environmental bacteria. Water may be decontaminated by reverse osmosis or UV radiation [82], although the former method alters the ion input provided by the water. UV radiation can also be used to decontaminate bedding and diet, as this method does not lead to nutrient degradation or moisture absorption [83].

To keep SPF conditions during an experiment, it is possible to rely on individually ventilated cages (IVC). Microenvironmental parameters, such as temperature, humidity [84], and light–dark cycle [85], which may lead to GM alteration, can be maintained by the IVC controller. However, noise and airflow conditioned by air filters may be potential stressors for animals [86]. The use of IVC is also limited to species with small body sizes. IVC are helpful to avoid the *cage effect*, which leads to GM homogenization and the loss of individual traits in a group of animals by mutual inoculation during contact with skin or feces (especially during coprophagy) [87]. The adopted strategy to deal with this is housing animals in individual cages, which discriminates against using herd animals with strong social needs. Nevertheless, social isolation and SPF conditions negatively affect the immune system’s maturation [88], potentially leading to GM dysbiosis.

Considering all of this, we reviewed the designs of animal experiments on the gut–kidney axis that were retrieved by the search (Table 1). Most of the studies were randomized (15 out of 23), performed on omnivorous colony animals (21 out of 23), or in a controlled environment (16 out of 23) with surgically induced CKD (17 out of 23). The weaknesses of the studies were a lack of information about diet composition, water source, sterilization methods, animal co-housing, and sterility of cages. Improving the reporting of GM-relevant study design parameters would improve the reproducibility and verifiability of results.

## 7. Conclusions

The co-evolution of GM and the host has led to highly complex and multifactorial relationships [89]. The holobiont, an organism composed of a host and a microbiome, combines a highly conserved human genome and a dynamic genome of the microbiome that can change rapidly in response to external factors, by a variety of means (increasing or reducing the number of particular microbes, acquisition of novel microbes, horizontal gene transfer, mutations) [90]. In health, the GM composition changes dynamically within alternative stable states with positive impacts on the host [91]. However, unfavorable factors may induce a critical transition of GM composition to dysbiosis, where GM metabolism is adverse to the host [91].

CKD models in rodents, but not in dogs, reproduce the elongated intestinal transit time observed in humans. Alterations of the gut barrier and endotoxemia have also been reproduced in these models. Interestingly, predominantly translocated taxa are similar in clinical and preclinical studies. During CKD, a shift towards a uremic toxin-producing GM and reduced production of beneficial SCFAs is observed. Despite GM compositional differences between human and animal models, the changes in GM metabolism in CKD have been well reproduced regardless of the approach used to induce CKD (diet or surgery). Different methods of inducing CKD in animal models are associated with different problems and limitations. The adenine-enriched diet causes crystal formation and tubulointerstitial damage. However, the altered content of diet itself may strongly influence the composition and metabolism of the GM. Moreover, the adenine-enriched diet adversely affects body weight and causes malnutrition. The biggest obstacle to studying the gut–kidney axis with this model appears to be the difficulty in having a control group able to distinguish the effects of the diet from those of CKD. Surgically induced CKD animal models (such as SNx and UUO) require the use of medication, which influenced the GM composition. In contrast to the diet-induced models, it is possible to perform sham surgery to control the effect of surgery and medications.

GM investigations have specific requirements that need to be taken into consideration when designing animal experiments. Preferably, omnivorous species with small body sizes that allow for housing in controlled conditions, ideally in IVC, should be used. Methods of disinfection of diet and bedding should be implemented. It is crucial to set up a control group that allows for distinguishing the GM effects of CKD from those linked to the CKD induction method. Thus, surgically induced CKD would be preferable to adenine-enriched diet-induced CKD for GM studies.

## Figures and Tables

**Figure 1 toxins-14-00626-f001:**
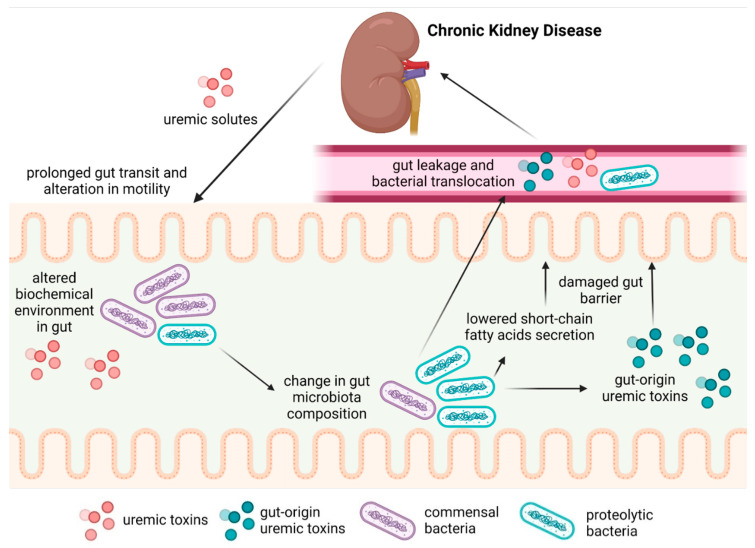
The gut–kidney axis during CKD.

**Table 1 toxins-14-00626-t001:** Main characteristics and results of CKD animal studies assessing gut and GI alterations retrieved by the search.

Study	Species	CKD Induction	Study Strengths	Study Weaknesses	Main Results
Lefebvre, (2001) [16]	Beagle male dogs	SNx	colony animals, confirmed CKD, individual cages	no randomization, carnivorous, no treated control	intestinal transit time↑
da Graça, (2015) [17]	Wistar male rats	SNx	randomization, omnivorous colony animals	no diet description	small intestinal transit↓
Wang, (2001) [18]	Sprague– Dawley male rats	SNx	omnivorous colony animals, confirmed CKD, individual cages	no randomization, no-sham control, tap water, no diet description	small intestinal transit↓ *(no change when fasting)*
Yu, (2018) [19]	Sprague– Dawley male rats	SNx	randomization, omnivorous colony animals	no diet description	small intestinal transit↓ inflammation, oxidative stress↑
Hoibian, (2018) [20]	C57Bl/6 JRj male mice	adenine diet	randomization, omnivorous colony animals	diet-induced CKD	total intestinal transit↓ colon motility↓
Gonzalez, (2019) [21]	rats	SNx	omnivorous, individual cages	no randomization, unknown rat line and sex, no diet description	gut barrier functions↑ diabetes↑ LPS↑
Huang, (2020) [22]	Balb/c male mice	SNx	randomization, blind testing, omnivorous colony animals	no housing description	GM alteration↑ gut injury↑ mRNA tight junction↓ gut permeability↑
Yang, (2019) [23]	C57BL/6 male mice	SNx	omnivorous colony animals, SPF environment	no randomization, no diet description	GM alteration↑ claudin-1↓ claudin-2↑ gut permeability↑ inflammation↑
Vaziri, (2013) [24]	Sprague– Dawley male rats	adenine diet	randomization, omnivorous colony animals	diet-induced CKD, no treated control	tight junctions↓ endotoxemia↑ inflammation, oxidative stress↑
Wang, (2012) [25]	Sprague– Dawley male rats	SNx	randomization, omnivorous colony animals	no housing and diet description	Intestinal permeability↑ bacterial translocation↑ inflammation↑
de Almeida Duarte, (2004) [26]	Wistar male rats	SNx	randomization, omnivorous colony animals	tap water	gut injury↑ bacterial translocation↑
Vaziri, (2013) [27]	Sprague– Dawley male rats	SNx	randomization, omnivorous colony animals	no diet description	tight junctions↓ oxidative stress↑
Yoshifuji, (2016) [28]	SHR male rats	SNx	randomization, omnivorous colony animals	no treated control, no housing and diet description	GM alteration↑ tight junctions↓ inflammation↑
Vaziri, (2012) [29]	Sprague– Dawley male rats	SNx adenine diet	randomization, omnivorous colony animals, two CKD models	no diet description	tight junctions↓ mRNA tight junctions↑ inflammation↑
Vaziri, (2020) [30]	Sprague– Dawley male rats	SNx	randomization, omnivorous colony animals		tight junctions↓
Lau, (2015) [31]	Sprague– Dawley male rats	SNx	randomization, omnivorous colony animals	no treated control, no housing and diet description	tight junctions↓ inflammation↑ oxidative stress↑
Vaziri, (2014) [32]	Sprague– Dawley male rats	adenine diet	randomization, omnivorous colony animals	diet-induced CKD, no treated control	tight junctions↓
Hung, (2018) [33]	ICR male mice	adenine diet	omnivorous colony animals, distilled water	diet-induced CKD, no treated control	GM alteration↑ gut barrier functions↓ inflammation↑ gut permeability↑
Chen, (2019) [34]	Sprague– Dawley male rats	UUO	randomization, omnivorous colony animals	no housing and diet description	GM alteration↑ tight junctions↓ uremic toxins↑ SCFAs↓ inflammation↑ oxidative stress↑
Hsu, (2020) [35]	Sprague– Dawley female rats	adenine diet	omnivorous colony animals	no randomization, diet-induced CKD	GM alteration↑ uremic toxins↑ SCFAs↓
Yano, (2014) [36]	Wistar male rats	SNx	omnivorous colony animals	no randomization, no housing and diet description	mRNA ABCG2↑ uricase activity *(stable)* uric acid *(stable)*
Ji, (2020) [37]	Sprague– Dawley male rats	SNx	randomization, omnivorous colony animals, SPF environment	no treated control, no diet description	GM alteration↑ gut injury↑ tight junctions↓ inflammation↑ LPS↑
Mishima, (2014) [38]	C57BL/6 male mice	adenine diet	randomization, omnivorous colony animals	diet-induced CKD, no housing description	GM alteration↑ gut barrier functions↓ uremic toxins↑

Abbreviations: ABCG2—ATP binding cassette subfamily G member 2; CKD—chronic kidney disease; GM—gut microbiota; LPS—lipopolysaccharides; mRNA—messenger ribonucleic acid; SCFAs—short-chain fatty acids; SNx—subtotal nephrectomy; SPF—specific-pathogen free; UUO—unilateral ureteral obstruction; ↑—increased; ↓—reduced.

**Table 2 toxins-14-00626-t002:** Summary of gut alterations observed in animal models of CKD induced by adenine-enriched diet or by surgery.

Characteristic	Adenine-Enriched Diet-Induced CKD	Surgery-Induced CKD
gut motility	reduced [20]	reduced [17,18,19] (increased for dogs [16])
leaky gut	not verified	induced [21,22,23]
bacterial translocation	induced [24]	induced [25,26]
tight junction	lower protein concentration [29,32,33,74]	lower protein concentration [21,22,23,28,29,30,31] structural alteration observed [22] inconclusive changes in tight junction mRNA [22,29]
transcellular transport in the intestine	not checked	increased expression of the ABCG2 urate transporter in the ileum [36]
gut microbiota	**Metabolic changes:** increased uremic toxins production, reduced SCFAs production [35] **Composition changes:** no similarity to the gut microbiota of CKD patients [35]	**Metabolic changes:** increased uremic toxins production, reduced SCFAs production [34,37] **Composition changes:** no similarity to the gut microbiota of CKD patients [34,37]
main disruptors of the intestinal microbiota	modified diet [38,68,69,70]	anesthetics, analgesics, antibiotics [71,72,73]

**Abbreviations:** ABCG2—ATP binding Cassette subfamily G member 2; CKD—chronic kidney disease; mRNA—messenger ribonucleic acid; SCFAs—short-chain fatty acids.

## Data Availability

Not applicable.

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
