# Peer review of "Gut–Kidney Axis Investigations in Animal Models of Chronic Kidney Disease"

_toxins, 2022, doi:10.3390/toxins14090626_

Round 1

Reviewer 1 Report (Previous Reviewer 3)

New figure 1 make this review attractive and easy to understand the author's opinion. Now, I think this review become acceptable.

Author Response

To whom it may concern,
I would like to thank you sincerely for your reviews and comments which allowed me to improve the manuscript. Figure 1 has been further modified to improve its resolution. 

Respectfully yours,
Piotr Bartochowski 

Reviewer 2 Report (Previous Reviewer 2)

Thank you for the opportunity to review the revised manuscript.

The comments raised by the reviewer are well addressed.

Author Response

To whom it may concern,
I would like to thank you sincerely for your reviews and comments which allowed me to improve the manuscript. 

Respectfully yours,
Piotr Bartochowski 

Reviewer 3 Report (New Reviewer)

As you highlighted in your article, altered gut microbiota can lead to important metabolic changes, especially in renal impaired patients, therefore, the correct assessment of gut-kidney axis is required for a better understanding of the pathophysiological mechanisms involved in CKD progression. Overall, your article was well written, and the information were based on updated data, but I have some few comments:

In the abstract, line 6 – “in the late stage” instead of “at the end stage”. In the abstract, line 9 – “are needed” instead of “is needed” (there are 2 subjects: “the ability of CKD” and “gut microbiota”).

- Lines 17-18 – maybe the word “regarding” should be deleted (similar for lines 76-77), and please consider to replace “In the review article” with “In this review” and “This article” with “The article” (at the beginning of the next sentence).

- Figure 1 seems not to have a sufficiently high resolution. Additionally, under the figure please add a note explaining the used abbreviations (even if they were explained in the manuscript). Similar for Table 1 – please explain the used abbreviations (as you did for Table 2).

- The paragraph written between lines 45 and 57 should be excluded from Introduction and renamed as Materials and Methods. Regarding lines 53-54 “We included studies on animal models of CKD only.” – please check the grammar (usually “only” is written next to the subject or verb). Line 55, it seems something is missing – “for eligibility by the author” instead of “for eligibility the author”, but please check it again.

- Line 100, as past tense was used, perhaps it should be “was” instead of “is” (“it was not”).

Please check again the English grammar in the entire manuscript.

Author Response

To whom it may concern,
I would like to thank you sincerely for your reviews and comments which allowed me to improve the manuscript. The indicated places have been corrected and the entire text has been checked and corrected for grammar and punctuation. The resolution of figure 1 has been corrected and the text has been modified to have no unclear abbreviations. Table 2 has received an explanation of the abbreviations. As advised, the Materials and Methods chapter has been separated.  

Respectfully yours,
Piotr Bartochowski 

This manuscript is a resubmission of an earlier submission. The following is a list of the peer review reports and author responses from that submission.

Round 1

Reviewer 1 Report

Comment To Authors

The manuscript is very interesting, the review takes into consideration various aspects associated with alterations in the intestinal microbiota and ckd, e.g. intestinal motility was different in patients with ckd, but also considers patients in esrd who need dialysis, also differentiating patients in hd and dp with actually different problems at the gastrointestinal level.

They also considered the different permeability in patients with ckd and esrd

Also the tight junction proteins and the expression of mRNA of tight junction has been analyzed in the various experimental models

Another important aspect, not clearly reported by the authors, is the biodiversity found in the various animal models studied compared to the controls

it is not clear, however, what the purpose of the authors and of this review is, in the sense that all the mechanisms associated with the alterations of the microbiota in the ckd have not been examined, certainly some but not all: The authors have certainly shown the elements for and against the use of some animal models over others

A dysbiotic gut microbiome may contribute to progression to CKD and CKD-related complications such as cardiovascular disease.

They did not mention this essential aspect for patients with ckd

CKD is associated with intestinal dysbiosis and alteration of the intestinal barrier, as reported by the authors.

Although the main cause of intestinal dysbiosis in CKD patients remains unknown, several hypotheses have been formulated, i.e. the increased transit of urea transformed into ammonia and ammonium hydroxide, the increased intestinal pH and altered intestinal barrier with bacterial translocation and subsequent endotoxemia, determining inflammation. Malnutrition, edema, fluid overload and intestinal wall congestion alter the intestinal blood flow and the colonic fecal transit, increasing intestinal barrier permeability. Low fiber diet, the main energy substrate for intestinal bacteria that produce SCFA, favors the increase of proteolytic bacteria. Frequent use of antibiotics and metabolic acidosis increase the catabolism of muscle proteins by promoting insulin resistance with an increase in microbiota species with proteolytic metabolism at the expense of saccharolytic metabolism.

The authors did not analyze these aspects in animal models

In particular, in ESRD, the intestinal bacterial flora is significantly altered. The most often reported changes in gut microbiota in CKD are related to the lower levels of Bifidobacteriaceae and Lactobacillaceae and higher levels of Enterobacteriaceae.

-Ramezani, A.; Raj, D.S. The gut microbiome, kidney disease, and targeted interventions. J. Am. Soc. Nephrol. 2014, 25, 657–670. -Lau, W.L.; Kalantar-Zadeh, K.; Vaziri, N.D. The Gut as a Source of Inflammation in Chronic Kidney Disease. Nephron 2015, 130, 92–98. -Anders, H.J.; Andersen, K.; Stecher, B. The intestinal microbiota, a leaky gut, and abnormal immunity in kidney disease. Kidney Int. 2013, 83, 1010–1016. - Shah, N.B.; Allegretti, A.S.; Nigwekar, S.U.; Kalim, S.; Zhao, S.; Lelouvier, B.; Servant, F.; Serena, G.; Thadhani, R.I.; Raj, D.S.; et al. Blood Microbiome Profile in CKD: A Pilot Study. Clin. J. Am. Soc. Nephrol. 2019, 14, 692–701.

SCFA, whose beneficial effects consist in regulating glycemic and lipid metabolism, maintaining intact the intestinal barrier and the intestinal pH, and regulating immune system and inflammatory response. CKD may determine dysbiosis, increasing inflammation and oxidative stress, favoring CKD-toxicity and disease progression as well as increasing cardiovascular risk in this population.

The cardiovascular risk associated with alterations in the microbiota has not been mentioned and is important considering that it is the leading cause of morbidity and mortality in patients with ckd.

Aspects concerning inflammation were also not treated

-Lau W.L., Savoj J., Nakata M.B., Vaziri N.D. Altered microbiome in chronic kidney disease: Systemic effects of gut-derived uremic toxins. Clin. Sci. 2018;132:509–522. doi: 10.1042/CS20171107. -Chung S., Barnes J.L., Astroth K.S. Gastrointestinal Microbiota in Patients with Chronic Kidney Disease: A Systematic Review. Adv. Nutr. 2019;10:888–901. doi: 10.1093/advances/nmz028.

No therapeutic approach was considered, have not suggested or evaluated the use of prebiotics, probiotics or combinations to show any benefits at the microbiota level.

Author Response

First Reviewer

Open Review

English language and style

( ) Extensive editing of English language and style required
( ) Moderate English changes required
(x) English language and style are fine/minor spell check required
( ) I don't feel qualified to judge about the English language and style

Comments and Suggestions for Authors

Comment To Authors

The manuscript is very interesting, the review takes into consideration various aspects associated with alterations in the intestinal microbiota and ckd, e.g. intestinal motility was different in patients with ckd, but also considers patients in esrd who need dialysis, also differentiating patients in hd and dp with actually different problems at the gastrointestinal level.

They also considered the different permeability in patients with ckd and esrd

Also the tight junction proteins and the expression of mRNA of tight junction has been analyzed in the various experimental models

  1. Another important aspect, not clearly reported by the authors, is the biodiversity found in the various animal models studied compared to the controls

Thank you for your comment. A sentence was added in the manuscript (lines 199-202):

“Changes in the diversity of the GM composition vary according to the method of induction (increases in SNx models[56], does not change when UUO[50], decreases when CKD is induced by adenine-enriched diet[51]), while it mostly decreases in CKD and ESRD patients[57].”.

  1. it is not clear, however, what the purpose of the authors and of this review is, in the sense that all the mechanisms associated with the alterations of the microbiota in the ckd have not been examined, certainly some but not all: The authors have certainly shown the elements for and against the use of some animal models over others

The article covers selected disorders of the renal-gastrointestinal axis. To clarify this, the text was modified and the selection of topics and references is now explained in the methods (lines 48-60):

“In order to test the validity of using animal models of CKD to study the alteration of the renal-gut axis, an electronic search was performed on material published from January 1, 1990, to April 1, 2022, in PubMed. We searched combinations of the terms: “chronic kidney disease”, “chronic renal disease”, “chronic renal failure”, “gut microbiota”, “gut microbiome”, “intestinal microbiota”, “tight junction”, “gut barrier”, “gut barrier transport”, “gut motility,” “intestinal transit,” “leaky gut”, “gut hyperpermeability”, “bacterial translocation” and “endotoxemia”. Search results were limited to English-language publications. In addition, references of selected studies were screened to identify eligible trials that were not found through the database searches. We included studies on animal models of CKD only. Each search result was independently reviewed for eligibility the author (P.B.). We excluded trials that reported: (1) no details on CKD induction in animal models; (2) no results on intestinal disturbances, and (3) for articles showing gut microbiota alterations: no indication of the effect of compositional change.”.

  1. A dysbiotic gut microbiome may contribute to progression to CKD and CKD-related complications such as cardiovascular disease. They did not mention this essential aspect for patients with ckd

We agree with the reviewer’s comment, and added the following sentence in the manuscript (lines 38-40):

”Alterations in the kidney-gut axis (Figure 1) may adversely affect the CKD development and increase the risk of comorbidities such as cardiovascular disease [15], which is the leading cause of mortality in patients with CKD.”.

  1. CKD is associated with intestinal dysbiosis and alteration of the intestinal barrier, as reported by the authors. Although the main cause of intestinal dysbiosis in CKD patients remains unknown, several hypotheses have been formulated, i.e. the increased transit of urea transformed into ammonia and ammonium hydroxide, the increased intestinal pH and altered intestinal barrier with bacterial translocation and subsequent endotoxemia, determining inflammation. Malnutrition, edema, fluid overload and intestinal wall congestion alter the intestinal blood flow and the colonic fecal transit, increasing intestinal barrier permeability. Low fiber diet, the main energy substrate for intestinal bacteria that produce SCFA, favors the increase of proteolytic bacteria. Frequent use of antibiotics and metabolic acidosis increase the catabolism of muscle proteins by promoting insulin resistance with an increase in microbiota species with proteolytic metabolism at the expense of saccharolytic metabolism.

To mention these highly relevant aspects, we developed the introduction as followed (lines 26-36):

“CKD patients have been reported to display reduced gut motility, increased intestinal permeability, bacterial overgrowth, bacterial translocation and gut-origin inflammation [4–6]. Although the main cause of intestinal dysbiosis in CKD patients remains unknown, there is evidence that low-fibre [7], low protein diet [8], the use of medications and treatments [9–11] and reduced renal function alter the biochemical milieu of the gut and modified gut microbiota (GM), the ecosystem of microorganisms inhabiting the intestine. The uremic syndrome also affects the gastrointestinal tract with manifestations of anorexia, nausea, ulcers, malnutrition, and protein-energy wasting [12]. CKD may favor the growth of uraemic toxin-producing bacteria. In parallel, the reduced presence of carbohydrate-fermenting bacteria leads to a decreased production of beneficial Short Chain Fatty Acids (SCFAs) [13], which are involved in the regulation of glycaemic and lipid metabolism, immune system response, blood pressure and gut barrier functions [14].”.

  1. The authors did not analyze these aspects in animal models

As observed by the reviewer, we reviewed specific aspects of the gut-kidney axis in animal models. To clarify our approach and the scope of the review, the text was modified and the selection of topics and references is now explained in the methods (lines 48-60).

  1. In particular, in ESRD, the intestinal bacterial flora is significantly altered. The most often reported changes in gut microbiota in CKD are related to the lower levels of Bifidobacteriaceae and Lactobacillaceae and higher levels of Enterobacteriaceae.

Due to interspecies differences and CKD origin, compositional changes in the gut microbiota are difficult to compare. Therefore, the focus was on similarities in the altered metabolism of the gut microbiota during CKD (chapter “Gut microbiota alterations in CKD”).

-Ramezani, A.; Raj, D.S. The gut microbiome, kidney disease, and targeted interventions. J. Am. Soc. Nephrol. 2014, 25, 657–670. -Lau, W.L.; Kalantar-Zadeh, K.; Vaziri, N.D. The Gut as a Source of Inflammation in Chronic Kidney Disease. Nephron 2015, 130, 92–98. -Anders, H.J.; Andersen, K.; Stecher, B. The intestinal microbiota, a leaky gut, and abnormal immunity in kidney disease. Kidney Int. 2013, 83, 1010–1016. - Shah, N.B.; Allegretti, A.S.; Nigwekar, S.U.; Kalim, S.; Zhao, S.; Lelouvier, B.; Servant, F.; Serena, G.; Thadhani, R.I.; Raj, D.S.; et al. Blood Microbiome Profile in CKD: A Pilot Study. Clin. J. Am. Soc. Nephrol. 2019, 14, 692–701.

  1. SCFA, whose beneficial effects consist in regulating glycemic and lipid metabolism, maintaining intact the intestinal barrier and the intestinal pH, and regulating immune system and inflammatory response. CKD may determine dysbiosis, increasing inflammation and oxidative stress, favoring CKD-toxicity and disease progression as well as increasing cardiovascular risk in this population.

In reply to your comment, the part of the introduction on the importance of SCFAs in homeostasis was developed (lines 34-38):

“The reduced abundance of carbohydrate-fermenting bacteria leads to a decreased production of beneficial Short Chain Fatty Acids (SCFAs) [13], which are involved in the regulation of glycaemic and lipid metabolism, immune system response, blood pressure and gut barrier functions [14].”.

  1. The cardiovascular risk associated with alterations in the microbiota has not been mentioned and is important considering that it is the leading cause of morbidity and mortality in patients with ckd.

Due to the relevance of this CKD comorbidity, this relationship is mentioned in a sentence: (lines 38-40):

“Alterations in the kidney-gut axis may adversely affect the CKD development and increase the risk of comorbidities such as cardiovascular disease [15], which is the leading cause of mortality in patients with CKD.”.

  1. Aspects concerning inflammation were also not treated

The article does not delve into the mechanisms of gut-origin inflammation. In animal models of CKD, systemic inflammation is detected as a part of CKD and bacterial translocation increase the level of inflammation markers in serum (as was mentioned in the lines 122-124):

“Bacterial DNA translocation was correlated with inflammation markers such as serum IL-6 and C-reactive protein [6,36].”

-Lau W.L., Savoj J., Nakata M.B., Vaziri N.D. Altered microbiome in chronic kidney disease: Systemic effects of gut-derived uremic toxins. Clin. Sci. 2018;132:509–522. doi: 10.1042/CS20171107. -Chung S., Barnes J.L., Astroth K.S. Gastrointestinal Microbiota in Patients with Chronic Kidney Disease: A Systematic Review. Adv. Nutr. 2019;10:888–901. doi: 10.1093/advances/nmz028.

  1. No therapeutic approach was considered, have not been suggested or evaluated the use of prebiotics, probiotics or combinations to show any benefits at the microbiota level.

The aim of the article was to demonstrate the ability of animal models to map the effect of chronic kidney disease on the renal-gut axis observed in CKD patients and to adapt the design of the preclinical experiments to obtain valid results. This forms the basis for the use of CKD therapies targeting the gut microbiota. The use of antibiotics, probiotics, prebiotics and bacterial metabolites is an extensive topic that has already been previously reviewed and was considered beyond the scope of our article.

For instance:

Koppe, Laetitia, Denise Mafra, and Denis Fouque. "Probiotics and chronic kidney disease." Kidney international 88.5 (2015): 958-966.

Ramezani, Ali, and Dominic S. Raj. "The gut microbiome, kidney disease, and targeted interventions." Journal of the American Society of Nephrology 25.4 (2014): 657-670.

Reviewer 2 Report

To the authors

Thank you for the opportunity to review this article.

The authors summarized the latest findings on the gut-kidney axis and animal models' role, making it an interesting topic. The manuscript is well organized, with very clear and concise explanations of the advantages and disadvantages of different animal models.

I have only two comments as described below.

<Specific comments>

In the conclusion section, the authors described that surgically induced CKD would be preferable to adenine-enriched diet-induced CKD for GM studies. In table 1, both SNx and UUO were presented as surgical models. For researchers interested in the gut-kidney axis, the authors should clearly describe the difference in effect for GM between these models.

It is preferable to add the table about summary for the differences in effect for GM between adenine-enriched diet-induced CKD and surgically induced CKD.

Author Response

Second Reviewer

Submission Date

30 June 2022

Date of this review

22 Jul 2022 19:52:38

Open Review

English language and style

( ) Extensive editing of English language and style required
( ) Moderate English changes required
( ) English language and style are fine/minor spell check required
(x) I don't feel qualified to judge about the English language and style

Comments and Suggestions for Authors

To the authors

Thank you for the opportunity to review this article.

The authors summarized the latest findings on the gut-kidney axis and animal models' role, making it an interesting topic. The manuscript is well organized, with very clear and concise explanations of the advantages and disadvantages of different animal models.

I have only two comments as described below.

<Specific comments>

  1. In the conclusion section, the authors described that surgically induced CKD would be preferable to adenine-enriched diet-induced CKD for GM studies. In table 1, both SNx and UUO were presented as surgical models. For researchers interested in the gut-kidney axis, the authors should clearly describe the difference in effect for GM between these models.

Thank you for your comment. Although both the model based on subtotal nephrectomy (SNx) and unilateral ureteral obstruction (UUO) are surgically induced models of CKD, the mechanism of induction and course of disease remain different. Because of the paucity of studies on the gut microbiota in CKD models (especially the UUO model), it is difficult to characterize the difference between these models.

Please find the changes made to the manuscript

  • (lines 350-360) “The different methods of inducing CKD in animal models are associated with distinct problems and limitation. Adenine-enriched diet causes crystal formation and tubulointerstitial damage. However, the toxicity of the diet adversely affects body weight and causes malnutrition. The biggest obstacle to studying the kidney-gut axis with this model appears to be the difficulty in creating a control group to distinguish between the effect of diet and the effect of CKD. The surgery induced CKD animal models (such as SNx and UUO) require usage of medicaments which influenced gut microbiota composition. In the contrast to dietary-induced models, it is possible to perform sham surgery to control the effect of medications.”.  
  1. It is preferable to add the table about summary for the differences in effect for GM between adenine-enriched diet-induced CKD and surgically induced CKD.

Characteristic

adenine-enriched diet-induced CKD

surgery-induced CKD

gut motility

Reduced

reduced (increased for dogs)

leaky gut

not checked

Induced

bacterial translocation

Induced

Induced

tight junction

lower protein concentration

lower protein concentration

structural alteration observed

inconclusive changes of tight junction mRNA

Transcellular transport in the intestine

not checked

increased expression of the ABCG2, urate transporter, in the ileum

Gut microbiota

Metabolic changes:

increased uremic toxins production, reduced SCFAs production

Composition changes:

no similarity to the gut microbiota of CKD patients

Metabolic changes:                increased uremic toxins production, reduced SCFAs production

Composition changes:

no similarity to the gut microbiota of CKD patients

Main disruptors of the intestinal microbiota

modified diet

anesthetics, analgesics, antibiotics

Reviewer 3 Report

This review summarized animal model to investigate association between kidney and gut. Recently gut-kidney axis is one of the important topics in kidney research field. Therefore, this review is worth publication. But, there were several points which should be improved.

Major

1.     To make illustrations which figure out the gut-kidney axis to make this review more attractive.

2.     Although authors wrote gut-kidney animal included rabbits, dogs, pigs, miniature pigs, sheep, non-human primates, table included rodents and one dog study. The table might be not perfect.

3.     Addition to #2, the methods to make literature search should be described in detail.

4.     Addition to #3, the one study(Alteration of the Intestinal Environment by Lubiprostone Is Associated with Amelioration of Adenine-Induced CKD.

Mishima E, Fukuda S, Shima H, Hirayama A, Akiyama Y, Takeuchi Y, Fukuda NN, Suzuki T, Suzuki C, Yuri A, Kikuchi K, Tomioka Y, Ito S, Soga T, Abe T.J Am Soc Nephrol. 2015 Aug;26(8):1787-94. doi: 10.1681/ASN.2014060530. Epub 2014 Dec 18. ) was described in text but this study is not included in table.

5.     The limitation and problems of these gut-kidney axis models should be summarized in some paragraph.

Author Response

Third Reviewer

Open Review

English language and style

( ) Extensive editing of English language and style required
( ) Moderate English changes required
(x) English language and style are fine/minor spell check required
( ) I don't feel qualified to judge about the English language and style

Comments and Suggestions for Authors

This review summarized animal model to investigate association between kidney and gut. Recently gut-kidney axis is one of the important topics in kidney research field. Therefore, this review is worth publication. But, there were several points which should be improved.

Major

  1. To make illustrations which figure out the gut-kidney axis to make this review more attractive.

To answer this comment, the Figure 1 was added in the revised manuscript.

  1. Although authors wrote gut-kidney animal included rabbits, dogs, pigs, miniature pigs, sheep, non-human primates, table included rodents and one dog study. The table might be not perfect.

Thank you for your comment. Various animal species are used in the development of animal models of CKD, but due to the specificity of research on the kidney-gut axis (described in the chapter 'Adaptation of CKD animal models to kidney-GM research'), almost exclusively rodents and, rarely, dogs are used. To avoid confusion in the sentence (in lines 61-63), only rodents and dogs were focused on:

“A variety of CKD animal models have been used to explore the pathophysiology of CKD and gut-specific alterations. Although rodent models are predominant, other used species including dogs [16].”

  1. Addition to #2, the methods to make literature search should be described in detail.

To clarify this, the text was modified and the selection of topics and references is now explained in the methods: (lines 48-60):

 “In order to test the validity of using animal models of CKD to study the alteration of the renal-gut axis, an electronic search was performed on material published from January 1, 1990, to April 1, 2022, in PubMed. We searched combinations of the terms: “chronic kidney disease”, “chronic renal disease”, “chronic renal failure”, “gut microbiota”, “gut microbiome”, “intestinal microbiota”, “tight junction”, “gut barrier”, “gut barrier transport”, “gut motility,” “intestinal transit,” “leaky gut”, “gut hyperpermeability”, “bacterial translocation” and “endotoxemia”. Search results were limited to English-language publications. In addition, references of selected studies were screened to identify eligible trials that were not found through the database searches. We included studies on animal models of CKD only. Each search result was independently reviewed for eligibility the author (P.B.). We excluded trials that reported: (1) no details on CKD induction in animal models; (2) no results on intestinal disturbances, and (3) for articles showing gut microbiota alterations: no indication of the effect of compositional change.”

  1. Addition to #3, the one study (Alteration of the Intestinal Environment by Lubiprostone Is Associated with Amelioration of Adenine-Induced CKD.

Mishima E, Fukuda S, Shima H, Hirayama A, Akiyama Y, Takeuchi Y, Fukuda NN, Suzuki T, Suzuki C, Yuri A, Kikuchi K, Tomioka Y, Ito S, Soga T, Abe T.J Am Soc Nephrol. 2015 Aug;26(8):1787-94. doi: 10.1681/ASN.2014060530. Epub 2014 Dec 18. ) was described in text but this study is not included in table.

Thank you for you remark. The table 2 was modified as followed:

Mishima, (2014)[64]

C57BL/6

male mice

adenine diet

randomization,

omnivorous

colony animals,

diet-induced CKD,

no housing description

GM alteration↑

gut barrier

 functions↓

uremic toxins ↑

  1. The limitation and problems of these gut-kidney axis models should be summarized in some paragraph.

As the answer the new table (Table 1.) was added in the manuscript, which summarized the observation of the effect of different models on kidney-gut axis.

Characteristic

adenine-enriched diet-induced CKD

surgery-induced CKD

gut motility

Reduced

reduced (increased for dogs)

leaky gut

not checked

Induced

bacterial translocation

Induced

Induced

tight junction

lower protein concentration

lower protein concentration

structural alteration observed

inconclusive changes of tight junction mRNA

Transcellular transport in the intestine

not checked

increased expression of the ABCG2, urate transporter, in the ileum

Gut microbiota

Metabolic changes:

increased uremic toxins production, reduced SCFAs production

Composition changes:

no similarity to the gut microbiota of CKD patients

Metabolic changes:                increased uremic toxins production, reduced SCFAs production

Composition changes:

no similarity to the gut microbiota of CKD patients

Main disruptors of the intestinal microbiota

modified diet

anesthetics, analgesics, antibiotics

 Furthermore, the text was modified as followed (lines 350-360))

 “The different methods of inducing CKD in animal models are associated with distinct problems and limitation. Adenine-enriched diet causes crystal formation and tubulointerstitial damage. However, the toxicity of the diet adversely affects body weight and causes malnutrition. The biggest obstacle to studying the kidney-gut axis with this model appears to be the difficulty in creating a control group to distinguish between the effect of diet and the effect of CKD. The surgery induced CKD animal models (such as SNx and UUO) require usage of medicaments which influenced gut microbiota composition. In the contrast to dietary-induced models, it is possible to perform sham surgery to control the effect of medications.”

Submission Date

30 June 2022

Date of this review

15 Jul 2022 07:53:44
